# Changing Our Food Habits One Bite at a Time: Exploring Young Flexitarians in a Country with a High Meat Intake

**DOI:** 10.3390/foods13203215

**Published:** 2024-10-10

**Authors:** Tanja Kamin, Andreja Vezovnik, Irena Bolko

**Affiliations:** Faculty of Social Sciences, University of Ljubljana, SI-1000 Ljubljana, Slovenia; andreja.vezovnik@fdv.uni-lj.si (A.V.); irena.bolko@fdv.uni-lj.si (I.B.)

**Keywords:** meat reduction, meat consumption, flexitarians, young adults, sustainable diet, behaviour change

## Abstract

Flexitarian diets have gained attention for their potential positive impact on human health and greenhouse gas emissions reduction. However, a critical question remains: Can the segment of flexitarians significantly contribute to necessary changes in our current unsustainable food systems? Our study addresses this gap by examining meat consumption habits among young adults (*n* = 1023) in a country with traditionally high meat intake. Furthermore, we focus on a subset of flexitarians (*n* = 286). Our findings reveal two distinct groups of flexitarians: ethical (*n*_1_ = 140) and utilitarian (*n*_2_ = 148). Utilitarian flexitarians exhibit a stronger preference for meat (*t*(284)= −15.180, *p* < 0.001), greater food neophobia (*t*(284) = −4.785, *p* < 0.001), and lower environmental awareness (*t*(284) = 7.486, *p* < 0.001) compared to Ethical flexitarians. The Ethical group, predominantly female (*χ^2^*(1) = 13.366, *p* < 0.001), demonstrates higher life satisfaction (*t*(284) = 5.485, *p* < 0.001), better health perceptions (*t*(284) = 5.127, *p* < 0.001), and stronger beliefs in reducing meat consumption (*t*(284) = −8.968, *p* < 0.001). Additionally, Ethical flexitarians hold more positive views on plant-based meat, perceiving it as healthier (*t*(284) = 4.326, *p* < 0.001) and more ethical (*t*(284) = 4.942, *p* < 0.001), and show a greater willingness to adopt it (*t*(284) = 7.623, *p* < 0.001). While both groups possess similar knowledge and willingness regarding cultured meat and insects, Ethical flexitarians view cultured meat more favourably (*t*(250.976) = 2.964, *p* = 0.003). Our study provides insights into the evolving trends of flexitarianism within Central and Eastern European countries, where research on meat consumption and flexitarianism is scarce. These insights hold value for promoting behaviour change toward reduced meat consumption for both health and environmental reasons. Additionally, they offer guidance to the food industry, including producers, sellers, and providers of meals in educational and employment facilities.

## 1. Introduction

The scientific evidence is clear: society needs to adopt a more plant-based diet to build healthy and sustainable food systems [1], protect the environment [2,3] and better address health, animal welfare, and food security concerns [4,5]. There is no common consensus on how to effectively achieve this other than to make changes on both the supply and the demand sides; however, the transition to more sustainable food systems cannot happen without strong support from consumers [6]. Although there is growing evidence on how reducing meat consumption can improve the environment, human health, and animal welfare, meat consumption remains high in high-income countries, exceeding dietary recommendations by 2–4 times, while it is increasing in developing countries [7,8]. The majority of people still consider their regular meat consumption to be natural, normal, necessary, and nice [9], so it is unlikely that large numbers of people would be willing to adopt a completely plant-based diet. A plausible, acceptable alternative is the normalisation of an emerging dietary trend based on a reduction in high meat consumption and a change in dietary preferences for animal foods. Such a dietary shift is consistent with the so-called flexitarian diet, which is receiving increasing research attention worldwide [4].

Quantitative studies on “flexitarians” have been booming in the past decade (see [10,11,12,13,14,15]). Although lately, the term “flexitarians” has regularly appeared in the scholarly literature, studies have also used the term “meat reducers” to address the same group of consumers [16,17,18,19,20,21,22,23]. In some studies, flexitarians were also referred to as “low meat consumers” [24], or “semi-vegetarians” [23]. A significant amount of studies focus on the transition towards flexitarianism or towards plant-based diets [23,25,26], and some explore transitions towards meat alternatives, such as cultured meat (see [27]), plant-based diets [28] or insects [29].

Flexitarianism refers to meat reduction on a part-time basis or a plant-based diet with low meat content [30]. A flexitarian “abstains from eating meat occasionally without abandoning meat totally—in contrast to vegetarians who follow a meat-free diet and vegans who follow a strict plant-based diet and abstain from consuming all animal-based foods” [4] (p. 531). Thus, a flexitarian diet has an ambiguous relationship with the limits of sustainability because the criteria of flexitarianism are not very precise: Some diets include all meats but with a clear (e.g., 1×/week, 1×/month, ‘meatless Monday’) or unclear (e.g., moderate consumption) limit, while others exclude red meat, etc. [2]. Flexitarianism seems to be a popular form of food-related behaviour change because it does not impose drastic and strict regulations for major protein changes. Despite the claim that flexitarianism is not a sufficiently radical behaviour change in light of climate change and the need for changes in food systems [31], some argue that completely eliminating meat at the population level is neither realistic nor healthy because, for a meat-free diet, people must be health conscious and have sufficient knowledge about nutrition [32]. In addition, some argue that flexitarian diets may have some of the best potential for reducing greenhouse gas emissions because they do not need to replace animal products [33]. It is argued that flexitarianism already represents a real segment of food consumers [34], but the question is, if this segment is big enough and devoted enough to contribute meaningfully to much-needed changes in current unsustainable food systems.

Despite the recent surge in research on flexitarianism [4], several critical questions remain unanswered. We lack clarity on the willingness of consumers to embrace flexitarian dietary changes, the extent to which flexitarianism is already practised, the various forms it takes, and the level of commitment among flexitarians [4,10,11,21]. Furthermore, existing studies predominantly focus on a handful of countries, including Australia, Canada, New Zealand, the USA, Denmark, the Netherlands, France, Portugal, and Switzerland [4]. In contrast, Central and Eastern European consumers (Slovenians, Slovakians and Lithuanians) exhibit less willingness to reduce red meat consumption compared to their counterparts in European Mediterranean, Northern, and Western European countries. Additionally, Central and Eastern European consumers are, on average, less inclined to replace meat with alternatives such as insects, cultured meat, plant-based options, or even traditional vegetarian dishes like vegetable stew [35].

To address regional disparities in willingness to adopt a flexitarian diet, further research is necessary to explore consumer behaviour in Central and Eastern European states and to identify demographic groups with the potential to lead the change towards reduced meat consumption. Our study focuses on the demand side (consumers) in Slovenia, where, as in many other European countries, meat consumption surpasses nutritional guidelines [36].

A significant proportion of the hope for transforming unsustainable food habits is placed on younger generations. There is evidence to suggest that young adults constitute the largest age group adopting a flexitarian diet [33,37]. This phenomenon is not unexpected, given that dietary patterns and habits evolve at a gradual pace [38]. These habits are intricately linked with an individual’s identity and sense of belonging to a social group [39] (p. 290).

The cohort of young adults represents a social group that is still in the process of forming their identities. As they transition to adulthood, they frequently relocate from their familial residences, initiating new lives in disparate environments. This transition requires a reassessment of existing habits and the formation of new ones, including dietary choices [40]. Nevertheless, research attention has only recently been directed towards this cohort of food consumers in the context of meat and non-meat consumption [33,41,42,43]. Our study contributes to filling this gap. It draws on data collected from a quasi-representative national sample of Slovenian young adults in order to identify distinct segments of young flexitarians. We discuss the significance of their new flexitarian foodways in driving much-needed changes within our current harmful and unsustainable food systems.

This study gains significance in light of recent political efforts in Slovenia to strategically reduce meat consumption for health and environmental reasons [36]. However, there remains a lack of robust evidence supporting informed decisions regarding changes in the demand and supply aspects of meat production and consumption. Previous research on this topic in Slovenia has been limited by small, opportunistic samples and narrow study scopes related to meat reduction [44].

This article employs a structured approach. Firstly, the methodology employed in this study will be outlined, together with insights into the design of the research. Subsequently, an examination of the study results is provided, with a focus on the conscious reduction of meat consumption among young people in Slovenia. We quantify the number of individuals who have already taken steps to reduce their meat intake and explore the willingness of others to do so in the near or slightly more distant future. By assessing these stages of change, we are able to gain valuable insights into the trajectory of Slovenian youth with regard to their meat consumption habits. Subsequently, a detailed analysis is presented of a specific subgroup: young individuals who meet the criteria for flexitarianism. The objective is to gain insight into the various types of young flexitarians in Slovenia and to evaluate their potential impact on unsustainable food systems. By advocating for reduced meat consumption and promoting a predominantly plant-based diet, these flexitarians are instrumental in driving positive change. In conclusion, our discussion situates our findings within the existing scholarly literature on flexitarianism and discusses the potential value of the study’s insights for those engaged in the promotion of behaviour change towards reduced meat consumption for health and environmental reasons.

## 2. Materials and Methods

### 2.1. Data Collection and Sample

For this cross-sectional study, we collected data using the JazVem online survey panel among Slovenian young adults aged 18–35 in December 2021. The survey panel was administered by Valicon, an online survey provider with a database that offers a quasi-representative sample of the Slovenian population. We employed quota sampling, stratified by gender, age groups, and region. We aimed to obtain the largest quasi-representative sample possible within our financial constraints, considering the studied phenomenon and sample sizes in previous similar studies. Our sample was sufficient to conduct the desired tests and measure the studied phenomena in the selected population for the first time. The questionnaire was reviewed with experts and the target population before it was administered to the final study sample. A total of 1023 respondents completed the questionnaire (*n* = 1023), with an average response time of approximately 26 min and 39 s.

### 2.2. Description of the Survey

The survey incorporated variables related to values, norms, and eating practices, which previous research has demonstrated to be closely associated with flexitarianism. In our analysis, we focused on a subsample of respondents, assessing their behaviour change using a series of questions aligned with the stage model of change [45]. Specifically, we inquired whether participants had consciously reduced their meat consumption before the survey or if they intended to do so within the next 30 days or six months.

Participants who reported reducing their meat consumption were included in the segmentation analysis, which relied on the following variables:Subjective social norms regarding meat consumption were assessed using three items rated on a 5-point scale (ranging from 1 = strongly disagree to 5 = strongly agree). After conducting a principal components and reliability analysis, we excluded one item. The final composite score was based on the remaining two items, explaining 83.9% of the variance and demonstrating a Cronbach’s alpha reliability of 0.81.Self-efficacy was measured using three items rated on a 5-point scale from 1 (strongly disagree) to 5 (strongly agree). The composite score accounted for 65.5% of the variance, with a Cronbach’s alpha reliability coefficient of 0.73.Meat attachment was assessed using the Meat Attachment Questionnaire [46]. This questionnaire comprises 22 items rated on a 5-point Likert scale, ranging from 1 (strongly disagree) to 5 (strongly agree). After conducting a principal component and reliability analysis, we excluded three items. Ultimately, we adopted a rotated two-component solution that accounted for 63.0% of the variance. The first component, labelled ‘meat-eating affinity’, reflects high scores among participants who believe that meat is natural, essential for a healthy diet, and that the right to consume meat should not be questioned. Conversely, the second component, labelled ‘meat-eating ignorance’, corresponds to low scores among participants who experience negative emotions and are reminded of the death and suffering of animals when consuming meat. The Cronbach’s alpha reliability coefficient for meat-eating affinity was 0.95, and for meat-eating ignorance, it was 0.87.Food neophobia was assessed using the Food Neophobia Scale [47]. This scale consists of ten items, rated on a 5-point scale ranging from 1 (strongly disagree) to 5 (strongly agree). After conducting a principal component analysis, three items were removed. The final one-component solution explained 52.8% of the variance. The reliability coefficient (Cronbach’s alpha) for this scale was 0.85.Environmental awareness was assessed with seven items on a 5-point scale from 1 (strongly disagree) to 5 (strongly agree). Due to the poor quality of the items resulting from the principal component analysis, we excluded three items, and the final solution with four items explained 54.7% of the variance. The reliability coefficient of Crombach’s alpha was 0.72.Life satisfaction was assessed with a question on a 10-point scale ranging from 1 (extremely dissatisfied) to 10 (extremely satisfied).Subjective health was rated on a 5-point scale from 1 (very poor) to 5 (very good).

In addition to the core dietary variables considered in the segmentation analysis, we extended our investigation to include the following variables:Dietary variables (food consumption and reasons for not eating (or eating less) meat). We measured the frequency of consumption of different animal products (red meat, poultry, milk, eggs, etc.), with a scale ranging from 1—“never consume” to 7—“3 times a day or more”. The responses were grouped into two categories for the analysis: The first category included the responses “never consumed” and “once a month or less”, while the second category included the remaining responses. Those who do not eat animal products or eat them less than once per month were asked about the reasons for not eating these foods (or eating less of these foods), which we grouped into three categories: “don’t like the taste or smell”, “reasons for environmental and animal suffering/exploitation” and “other reasons” (which also included responses about health effects, weight control and religious reasons).Classic sociodemographic characteristics: gender (female, male or other), educational level (answers were aggregated into two categories: “primary/vocational or high school” and “college or university degree”), relationship (being in a relationship or not), living conditions (living in a village, in the suburbs, in a town or in a city), self-assessment of financial struggles (on a 6-point scale, ranging from 1 (very difficult to make through the month) to 6 (very easy to get to make through the month), and self-assessment of social status (on a 10-point scale, ranging from 1 (top of the society) to 10 (bottom of the society).Attitudes towards reducing meat consumption were measured using the ambivalence scale [48]. Respondents were asked, “What do you think of the idea of reducing meat consumption to once a week?” Respondents chose between the following items on a semantic-differential scale: bad-good, harmful-beneficial, unpleasant-pleasant, unsatisfactory-satisfactory, stupid-smart and impossible-possible. Responses were converted into a 5-point scale.Attitudes towards alternative (ALT) proteins (cultured meat, plant-based meat, and insects). The participants were asked whether they had ever replaced meat with ALT proteins. They were then asked whether they knew what cultured meat, plant-based meat and foods containing insects were and whether they would be willing to try these foods. Attitudes towards meat and ALT proteins (plant-based, cultured meat, insects) were measured using a series of affirmative responses. Respondents had to choose between opposing statements about meat and ALT proteins. The instrument was adapted from [49]. Familiarity and experience with ALT proteins were measured by asking respondents about their knowledge of ALT proteins and their direct experience with consumption [50,51].Trusting various information sources: participants responded on a 5-point scale, ranging from 1 (do not trust at all) to 5 (completely trust), indicating the extent to which they trust various information sources. After conducting a principal component analysis, we grouped their answers into four categories: authority sources (including scientists, science institutions, health workers, and health organisations), official sources (such as governmental institutions and media), unofficial sources (including NGOs, web forums, and social platforms), and personal sources (family and friends).

### 2.3. Data Analysis

The primary objective of our analysis was to investigate a subset of respondents (N = 286) who reported reducing their meat consumption in the past year (excluding fish). Our focus was on identifying distinct segments based on their dietary habits and general attitudes toward food, with a specific emphasis on meat-related behaviours.

To achieve this, we employed a segmentation approach guided by the core dietary variables previously discussed in the introduction. These variables have both theoretical and empirical significance in existing research. To ensure internal consistency, we conducted a principal component analysis and calculated Cronbach’s Alpha reliability coefficients. The selected variables included meat attachment, social norms, self-efficacy, food neophobia, environmental awareness, life satisfaction, and subjective health assessment. While the first five variables were incorporated as standardised mean scale scores, the latter two were treated as single scores.

We employed Ward’s hierarchical clustering method without pre-established conditions regarding the number, size, or content of the groups. Similarity between units was assessed using Euclidean distance. The optimal number of clusters was determined through an analysis of dendrogram levels and guided by theoretical and content considerations [52]. Our goal was to create clusters where units within each group were as similar as possible in terms of the segmented variables while clusters themselves were as dissimilar as possible.

Before analysis, all variables were standardised. We conducted the clustering using IBM SPSS Statistics 27. Notably, all these variables were either good ordinal measures or could be transformed into such measures.

Based on the results of hierarchical clustering, we identified two distinct groups corresponding to the primary types of young flexitarians (*n*_1_ = 140, *n*_2_ = 148). Group membership for each respondent was determined using the K-means clustering procedure. To explore the characteristics of these groups, we performed independent sample *t*-tests for numeric variables, calculating Cohen’s *d* as a measure of effect size. For categorical variables, we employed chi-square tests with phi-coefficient as the effect size measure.

### 2.4. Ethical Considerations

Our study was conducted in accordance with the Code of Ethics for Researchers at the University of Ljubljana [53]. The study was explained to participants through an online questionnaire. Participants were informed that they would participate in the survey using their personal communication technologies and that all data would be anonymised and only reported in the aggregate. All participants acknowledged an informed consent statement in order to participate in the survey and were able to withdraw from the survey at any time without giving a reason. The dataset was anonymised, ensuring that no identifiable data of participants were included. The study has received consent from the Ethical Committee of The University of Ljubljana, Faculty of Social Sciences (No. 801-2024-011/TD).

## 3. Results

### 3.1. Intention of Young Adults in Slovenia to Reduce Their Meat Consumption

In the sample of young adults (*n* = 1023), less than one-third (28.0%) reported a conscious reduction in their meat consumption over the preceding year. The majority of respondents (72.0%) indicated that they had not reduced their meat intake during the specified period.

Among those who had not yet reduced their meat consumption (*n* = 737), 1.9% indicated plans to do so in the near or somewhat distant future. Furthermore, 21.4% of respondents indicated that they were undecided about reducing their meat consumption within the next six months, while 14.9% stated that they were uncertain about doing so within the next 30 days. These findings indicate that a proportion of respondents were receptive to the notion of reducing their meat intake despite having not previously contemplated this possibility.

Nevertheless, the majority of respondents (83.0% for the near future and 76.7% for the somewhat distant future) indicated that they had no intention of reducing their meat consumption.

### 3.2. Describing Young Flexitarians in Slovenia

In this study, we examined respondents who reported a reduction in their meat consumption (excluding fish) in the year prior to the survey. In accordance with the statistical and substantive criteria delineated in the methodology section, two distinct groups were identified: (1) Ethical flexitarians (*n*_1_ = 140) and (2) Utilitarian flexitarians (*n*_2_ = 148).

Statistically significant differences were observed between the two groups (see Table 1). Utilitarian flexitarians demonstrated a greater affinity for meat consumption and exhibited a greater level of meat-related ignorance compared to Ethical flexitarians. Furthermore, Utilitarian flexitarians exhibited higher scores on the food neophobia scale, whereas Ethical flexitarians demonstrated heightened environmental awareness, stronger subjective social norms pertaining to meat consumption, and enhanced self-efficacy in reducing meat intake. Furthermore, Ethical flexitarians demonstrated higher levels of life satisfaction and more positive perceptions of their health than Utilitarian flexitarians.

With regard to demographic variables, the data revealed a gender disparity between the groups. The Ethical flexitarians were predominantly female. No significant differences were identified in other demographic variables (see Table 2). Additionally, Ethical flexitarians reported superior financial well-being compared to Utilitarian flexitarians. Moreover, the two groups demonstrated disparate levels of trust in information sources. Those who identified as Ethical flexitarians demonstrated a greater propensity to place trust in medical professionals, healthcare institutions, educators, academic figures, and the scientific community. Nevertheless, no significant discrepancies were identified between the groups with respect to their trust in government institutions, media outlets, online forums, social media platforms, family members, or friends.

The Ethical flexitarians reported consuming less meat (beef, pork and poultry) and animal products (eggs and dairy products) than the Utilitarian flexitarians (see Table 3). The motives for reduced meat consumption differed between the two groups. Ethical flexitarians cited concerns related to animal exploitation and the negative environmental impact of meat consumption, whereas Utilitarian flexitarians more frequently mentioned sensory reasons (disliking taste or smell) or other factors, such as high meat prices, unfamiliarity with certain meats, or health concerns (see Table 4).

The ethical flexitarians demonstrated a greater proclivity towards the notion of reducing their meat consumption to a maximum of once a week, in comparison to the utilitarian flexitarians (see Table 5). The latter group tended to view this idea in a negative light, perceiving it as harmful, unpleasant, unsatisfactory, stupid, and even implausible.

Among Ethical flexitarians, 112 (80.0%) reported replacing meat with alternative proteins (ALT), compared to 85 (58.2%) of Utilitarian flexitarians. This difference is statistically significant (Fisher-Freeman-Halton Exact Test = 16.1, *p* < 0.001, *Phi* = 0.236). Nevertheless, a more detailed examination of specific ALT proteins reveals a less pronounced differentiation between the groups (see Table 6). For example, both groups demonstrated comparable levels of knowledge regarding cultured meat and insects and exhibited similar levels of willingness to consume these foods. Approximately half of the participants indicated a willingness to sample cultured meat, while less than a quarter expressed a similar interest in insects. A mere handful of participants indicated that they had already sampled insects.

Ethical flexitarians held stronger beliefs about plant-based meat compared to Utilitarian flexitarians. Specifically, they perceived plant-based meat as healthier, more natural, better for the environment and animals, more ethical, more attractive, tastier, safer, more nutritious, more necessary, better, and more available compared to utilitarian flexitarians (Table 7). Furthermore, Ethical flexitarians were more inclined to consume plant-based meat when it became more widely available in shops and restaurants (*M* = 3.89, *SD* = 1.14) than Utilitarian flexitarians (*M* = 2.92, *SD* = 1.01). This difference is statistically significant (*t*(284) = 7.623, *p* < 0.001, Cohen’s *d* = 0.902).

While Ethical and Utilitarian flexitarians exhibited no significant divergence in their knowledge of cultured meat or their willingness to sample it, notable distinctions emerged in their beliefs about cultured meat (Table 8). Ethical Flexitarians held more favourable perceptions of cultured meat. They perceived it to be a more environmentally and animal-friendly alternative, safer to consume, more nutritious, tastier, and overall superior. Nevertheless, no significant differences were observed between the groups with regard to their perceptions of cultured meat’s healthiness, naturalness, ethical implications, attractiveness, taste, expense, or necessity. Furthermore, both groups exhibited comparable perceptions regarding the probability of consuming cultured meat in the event of its widespread availability in retail and dining establishments. The mean rating for ethical flexitarians was 3.10 (*SD* = 1.23), while that for Utilitarian flexitarians was 3.01 (*SD* = 0.95). The Welch’s t-test yielded a value of 0.661 (*df* = 261.94), indicating no statistically significant difference (*p* = 0.509). The effect size was calculated to be 0.079, which is considered to be a negligible difference.

It would appear that both Ethical and Utilitarian flexitarians exhibit comparable attitudes with regard to the consumption of insects. As evidenced in Table 9, there were no statistically significant differences between the two groups in their perceptions of edible insects. Neither group considered insects to be particularly healthy, natural, environmentally friendly, ethical, attractive, or tasty. Furthermore, both groups demonstrated a lack of enthusiasm for consuming insects.

## 4. Discussion

### 4.1. Young Adults’ Intentions to Reduce Meat Intake in Slovenia

In Slovenia, the relationship between the dietary habits of young adults and meat consumption is characterised by a certain degree of complexity. While some are actively considering or open to reducing their meat intake, a significant portion of young adults continue to adhere to meat-dependent dietary patterns that are prevalent in the general population. It is noteworthy that this level of adherence exceeds the EU average for meat consumption by approximately 30% [54]. The present study surveyed respondents on their intentions regarding meat consumption. The majority of respondents indicated that they had no intention of reducing their meat intake in the near future (83.0%) or even in the somewhat distant future (76.7%), which is to some degree in concordance with a recent study in Slovenia completed on a more opportunistic sample of the adult population [44]. An interpretation of these findings through the lens of the Stages of Change Model [45] reveals that the majority of respondents remain in the pre-contemplation stage regarding the reduction of their meat consumption. This model outlines various stages individuals typically progress through during behavioural change, including pre-contemplation, contemplation, preparation, action, and maintenance. This indicates that they may be unaware of the necessity to reduce their meat consumption or do not consider it a priority. Some individuals may even express opposition to the notion of reducing meat consumption entirely. An examination of the historical context reveals the existence of this resistance. For decades, meat consumption has been encouraged and taken for granted in Slovenia. Only recently has it begun to be subjected to critical examination from both a health and an environmental perspective [36]. Consequently, the Slovenian population has demonstrated a reluctance to alter their meat consumption habits. In 2019, approximately 24.1% of Slovenians reported consuming meat and/or meat products at least once per day [55]. Over the past two decades, the mean annual per capita consumption of meat has remained constant at approximately 90 kg [54].

In our survey, a small but noteworthy percentage (1.9%) of respondents indicated their intention to reduce meat intake either in the near or somewhat distant future. This places them in the contemplation and/or preparation stage of behaviour change [45]. Furthermore, 21.4% of participants indicated that they were undecided about reducing their meat consumption within the next six months, while 14.9% were uncertain about doing so within the next 30 days. These findings indicate that a notable proportion of respondents are receptive to the notion of reducing their meat intake, even if they have not previously actively pursued this goal. An application of the Stages of Change Model [45] allows for the categorisation of these respondents as being in the pre-contemplation and/or contemplation stage. In this stage, individuals may be unaware of the necessity to reduce meat consumption, may not perceive it as relevant, or may not be prepared to make a decision at this time. Nevertheless, their receptivity to the concept indicates a prospective transformation in attitudes towards meat consumption, particularly when contextualised within Slovenia’s historical tradition of promoting meat consumption.

In the course of our study, 28% of respondents indicated that they had taken action to reduce their meat intake in the year preceding the survey and are thus situated within the action and/or maintenance stage of behaviour change [45]. The term ‘action’ pertains to a recent change, whereas ‘maintenance’ denotes a longer period during which the novel behaviour has been sustained, typically exceeding six months. It is noteworthy that the proportion of flexitarians among young adults is marginally higher than that observed in the general Slovenian population [55], which contradicts previous claims that middle-aged and elderly individuals are more likely than younger adults to report meat reduction behaviours [44]. Perhaps this discrepancy derives from the design of that study, which was based on an opportunistic sample that does not allow generalisation on the whole population [44]. In another study from 2019, approximately 20% of the adult population (aged 18–75) in Slovenia could be considered flexitarians, consuming meat and meat products 1–3 times per year, less than once per week, or 1–2 times per week [55]. The motives of older adults for excluding meat from their daily diet could be economically driven due to higher prices of meat, as another study suggests [38]. While comparisons between data from different samples and studies are inherently challenging, the evidence suggests that young adults in Slovenia may be more receptive to reducing meat consumption than the overall population. This trend is also reflected in the percentage of vegans and vegetarians. The proportion of young adults in our study who defined themselves as vegetarians or vegans was 3%, whereas the proportion of non-meat eaters in the adult population of Slovenia ranged from 1.4% to 1.6% [56]. This finding is consistent with the findings from Denmark, indicating that young adults were more likely to contemplate reducing or even abstaining from meat consumption compared to older age groups. [21]

### 4.2. Heterogeneity among Young Adult Flexitarians: Implications for Sustainable Food Systems

As has been demonstrated in previous research examining the diverse nature of flexitarians (e.g., [11,20,25,57]), our findings revealed that the category of flexitarians among young adults in Slovenia is far from homogeneous. The study identified two distinct types of flexitarians:(1)Ethical flexitarians: These individuals reduce their meat consumption primarily due to ethical concerns related to animal welfare and environmental impact. They demonstrate a tendency to place trust in institutions, health professionals, and scientists. Ethical flexitarians exhibit openness to novel foods and dietary practices. Individuals in this group tend to demonstrate openness to novel foods and dietary practices. This group is more likely to comprise women, individuals with superior subjective health, higher life satisfaction, and greater financial well-being. Subjective social norms and perceived behavioural control exert a significant influence on their decisions to reduce meat consumption. Moreover, in comparison to the other group, they exhibit greater self-efficacy in reducing their meat intake and consume less meat overall.(2)Utilitarian flexitarians: This group displays a stronger attachment to meat and exhibits greater resistance to adopting new dietary practices. This group displays a greater tendency towards food neophobia. The decision to reduce meat consumption is predominantly driven by factors pertaining to convenience and efficiency, in addition to sensory attributes such as taste and odour. Moreover, financial considerations also inform their decisions to reduce meat intake. In contrast to those who adopt a flexitarian diet for primarily ethical reasons, utilitarian flexitarians tend to exhibit less concern for environmental and animal welfare issues and consume a greater quantity of meat.

In alignment with extant research [16], our study uncovers notable discrepancies among flexitarians with regard to attitudes, beliefs, motives, sociodemographic characteristics, and psychological variables. Previous research has categorised flexitarians based on a number of different criteria, including meat consumption frequency [11,20,25,57], willingness to reduce meat intake [18], and predictors of flexitarianism [57]. These studies highlight the necessity of elucidating motives and demographic variables in order to differentiate between the various typologies of flexitarians.

In terms of motives, the two distinct groups of young flexitarians in Slovenia align with the categorisations identified in previous research elsewhere. These categorisations have classified flexitarians into two main groups: (1) Personal and Intrinsic Motives Group: This group is comprised of individuals who place a premium on health, price, and taste. The motives of this group are more personal and intrinsic [11]. This group can be likened to our Utilitarian flexitarians; (2) and Extrinsic and Ethical Motives Group: This group is comprised of individuals whose motives are more extrinsic and ethical in nature. These individuals place a premium on animal welfare and environmental protection [10,22,33]. This group corresponds to the Ethical flexitarians. In a qualitative study, [33] a similar categorisation of young flexitarians to ours was observed. The group resembling our Utilitarian flexitarians was characterised by individual motives, including health, food variety, price, and reduced social discomfort. In contrast, the group resembling our Ethical flexitarians was motivated by altruistic concerns related to the environment and ethics.

Our study reveals that, as has been demonstrated in previous research [22,24,57], both groups of flexitarians exhibit a combination of ethical and utilitarian motivations for reducing their consumption of meat. These shared motivations include concerns related to health, taste, environmental impact, and animal welfare. In addition to the motivating factors mentioned earlier, some identified other factors influencing meat reduction, including cost and weight control [19]. Our study specifically highlights the cost of meat as a significant motivator for reducing meat consumption among Utilitarian flexitarians.

Those who adopt an ethical approach to their diet are more likely to be driven by extrinsic motivations, such as concerns for the environment and animal welfare when considering a reduction in meat consumption. These individuals consume less meat than their utilitarian counterparts and indicate a willingness to further reduce their meat intake. From this perspective, our study highlights the potential influence of ethical flexitarians in promoting positive transformation within food systems. Another study revealed that animal welfare concerns and environmental concerns, which were both significant motivators for reducing meat intake among Ethical flexitarians, were predictive of subsequent adoption of a vegetarian diet [57,58].

Utilitarian flexitarians, on the other hand, maintain a stronger attachment to meat, emphasising individual benefits associated with reduced meat consumption. Convenience and sensory factors play a pivotal role in their dietary choices. Unlike Ethical flexitarians, they pay less attention to environmental and animal welfare considerations. Our findings align with previous studies [5,10,11,17] revealing that the group of flexitarians who significantly reduced meat consumption prioritised environmental and animal welfare concerns over other, more utilitarian motives such as health, price, taste, and safety.

A number of studies have indicated that individuals who limit their meat consumption due to environmental and animal welfare concerns are typically female [10,22,59]. Our study also corroborates this trend, indicating that women are more likely to adhere to ethical flexitarian principles, whereas men tend to espouse utilitarian flexitarianism. It is noteworthy that while some studies have identified differences between urban and rural meat reducers, our study based on young adults does not corroborate this distinction.

Furthermore, our study illuminates the relationship between flexitarianism and socioeconomic position. It is notable that socioeconomic factors play a pivotal role in the formation of dietary behaviours, including those related to meat consumption. This finding corroborates existing research, which highlights the impact of financial resources and education on an individual’s capacity to modify their dietary choices. In cultures where meat-heavy culinary traditions are the norm, the adoption of diets that significantly reduce or exclude meat requires substantial dietary shifts. These changes are closely intertwined with consumers’ attitudes towards meat consumption, which are, in turn, strongly influenced by their socioeconomic status [38,60]. The findings of our study indicate that Utilitarian flexitarians exhibit heightened price sensitivity and report lower levels of life satisfaction and financial well-being when compared to Ethical flexitarians. Moreover, they seem less inclined to curtail their meat consumption further. This leads to the formulation of an intriguing hypothesis: It may be hypothesised that the financial sensitivity of this group of flexitarians may influence the quantity of meat consumed. It seems reasonable to posit that an improvement in their financial status or lower price of meat might lead to an increase in meat consumption. From this perspective, Utilitarian flexitarians are not a reliable indicator of the flexitarian movement’s potential to address unsustainable food systems.

### 4.3. Meat Replacement Preferences among Flexitarians

Additionally, our study illuminates the preferred meat replacement options among those who adhere to a flexitarian diet. The findings revealed that the majority of flexitarians surveyed opted to substitute meat with other animal-based proteins, such as dairy and eggs. The next most frequently selected meat replacements are plant-based alternatives, particularly pulses and other plant protein sources, such as oats. These findings are consistent with those of previous studies [24,61], which similarly observed that meat is most commonly replaced by other animal-derived products (such as fish, eggs, and cheese) and subsequently by plant-based options (such as instant meat substitutes, legumes, nuts, and soy products). In one study [62], the most frequently consumed plant-based alternatives were grains, nuts, seeds, legumes, and tofu. Another study [63] emphasised that highly processed meat analogues that closely resemble meat have the greatest potential for successfully replacing traditional meat. Nevertheless, flexitarians demonstrate a relatively low level of interest in alternative proteins, particularly novel options such as algae, cultured meat, insects, and mycoprotein [64]. Alternatively, they tend to favour more familiar options, such as peas, lentils and whole grains, as meat substitutes [64].

In our sample, 80% of ethical flexitarians reported substituting meat with one of the alternatives (whether animal-sourced or not), while a significantly lower percentage of utilitarian flexitarians (58%) did the same. This discrepancy is likely attributable to the fact that ethical flexitarians typically consume less meat than their utilitarian counterparts, thereby increasing their reliance on meat replacements. A comparable pattern was observed by others [65] among omnivores, flexitarians, and vegans. Those who expressed a more favourable view and greater willingness towards plant-based meat were predominantly vegans, while omnivores demonstrated the least interest. Flexitarians exhibited relatively low interest in meat analogues (such as vegan sausages, burgers, and nuggets), falling between the two aforementioned categories.

It is noteworthy that this finding is consistent with the European consumer preferences study [35], which indicates that Slovenians, for instance, are more inclined to substitute meat with traditional vegetarian foods (e.g., vegetable stew) than with plant-based meat alternatives, even if the latter are free of genetically modified organisms (GMOs). Nevertheless, individuals who prioritise ethical considerations tend to demonstrate a greater inclination towards a broader spectrum of meat alternatives [66]. In particular, the Ethical flexitarians in our study expressed openness to plant-based meat alternatives and held a more positive attitude toward them than the utilitarian group. Furthermore, they anticipate an increase in their consumption of plant-based meat alternatives as these options become more widely available on the market.

### 4.4. Attitudes toward Novel Protein Alternatives among Flexitarians

In the context of novel foods, flexitarians have been observed to exhibit hesitancy and even aversion towards certain protein alternatives [64]. In our sample, those with an ethical flexitarian perspective evinced a more favourable disposition towards cultured meat than those with a utilitarian flexitarian perspective. It is noteworthy that the willingness to try and incorporate cultured meat into one’s diet becomes evident, particularly when this alternative is widely available and consumed. Nevertheless, in comparison to plant-based proteins, interest in cultured meat remains relatively limited. This trend is consistent with the findings of other studies [65,67], which demonstrate that cultured meat encounters resistance across all dietary groups, including flexitarians.

Nevertheless, a notable discrepancy in attitudes towards cultured meat is evident between the two groups within our sample. Those who adopt an ethical flexitarian approach perceive cultured meat as a beneficial means of improving animal welfare and environmental sustainability, which aligns with their primary motivation for reducing meat consumption.

It is noteworthy that attitudes towards insects are negative in both groups of flexitarians. This finding is consistent with those of previous studies [35,65,67], which collectively indicate that insects are the least appealing alternative across all dietary groups.

### 4.5. Strengths and Limitations

The value of this study lies in its comprehensive approach to understanding the factors influencing the reduction in meat consumption. By analysing a quasi-representative sample of young people in Slovenia, the study assesses the type, size, and potential of young consumers in contributing to the transformation of food systems toward greater sustainability. It is noteworthy that the research addresses a significant gap in the existing literature by focusing on a specific population group: young people. This is a valuable approach to understanding future food trends. Furthermore, the study makes a valuable contribution to the limited research on flexitarianism in Central and Eastern European countries with high meat consumption.

However, it should be noted that the study is not without limitations. Firstly, the use of an online survey panel may have resulted in an over-representation of specific demographic groups among young adults with internet access and digital literacy, potentially affecting the representativeness of the results. Secondly, the use of self-reported behaviour introduces a potential source of bias. Thirdly, the cross-sectional design precludes the identification of cause-and-effect relationships. Fourthly, there are methodological concerns regarding the limited variance explained by a single extracted component, especially in the contexts of Food Neophobia and Environmental Awareness. Unfortunately, our findings did not support the extraction of multiple components that would substantially enhance the total explained variance.

Finally, it should be noted that the data is limited to the social, cultural, and political context of Slovenia, which has a strong tradition of meat-based cuisine.

## 5. Conclusions

Even young people today, who have the opportunity to choose between different dietary patterns, often continue to adhere to the established dietary practices that they learned in their primary social environment. The majority of young adults in Slovenia are not necessarily aware of the correlation between health and environmental issues associated with meat consumption, may deny the existence of problems related to meat consumption, and/or are not motivated to address the issue by modifying their dietary habits and reducing their meat consumption. A considerable number of respondents nevertheless expressed openness to the idea of reducing their meat intake and indicated a potential for a future shift in attitudes towards meat consumption. A noteworthy proportion of young adults indicated their intention to reduce their meat intake.

Given Slovenia’s historical tradition of promoting meat consumption, it is crucial to raise awareness of the environmental footprint related to the production and consumption of meat in the context of the country’s extensive meat consumption culture. Interventions could concentrate on incremental alterations, underscoring the significance of incremental advancements in the direction of meat reduction. For example, the promotion of “Meatless Mondays” or the encouragement of plant-based substitutions in familiar dishes may be effective strategies. Proper intervention strategies should facilitate the accessibility of plant-based meals in comparison to those that include meat nudge behaviours in grocery shops and canteens in educational or employment facilities. Moreover, social norms (instrumentalised via peer-led initiatives, social media campaigns, and community events) may be efficacious in emphasising that a considerable proportion of one’s social circle is already reducing meat consumption. This may encourage flexitarians to maintain their behaviour and persuade others to follow suit.

The recognition of several social, psychological, environmental, and practical factors could facilitate flexitarians to maintain their changed behaviour and enhance its impact. The availability (and affordability) of plant-based alternatives is a significant factor. Those who adopt a flexitarian diet for environmental or ethical reasons may be more likely to maintain their dietary change, less likely to revert to a diet high in meat, and more likely to continue reducing their meat intake. Exposure to educational campaigns or documentaries has the potential to reinforce this commitment. In contrast, Utilitarian flexitarians require social recognition of their dietary choices and additional encouragement to maintain or reduce their meat consumption. Furthermore, they benefit from the convenience and affordability of plant-based and alternative proteins.

Addressing food neophobia and resistance to trying new foods is of paramount importance for individuals across the behavioural spectrum, including flexitarians. It would be beneficial for interventions to educate young adults about alternative protein sources, cooking techniques, and the variety of plant-based options that are currently available.

## Figures and Tables

**Table 1 foods-13-03215-t001:** Descriptive Statistics with *t*-test Comparisons between Ethical flexitarians (*n*_1_ = 140) and Utilitarian flexitarians (*n*_2_ =146) for Segmentation variables.

Segmentation Variable	Ethical Flexitarians *M* (*SD*)	Utilitarian Flexitarians *M* (*SD*)	*t* (*df*), *p*	Cohen’s *d*
Meat attachment—affinity	2.05 (0.69)	3.21 (0.61)	−15.180 (284), <0.001	1.796
Meat attachment—ignorance	2.83 (0.92)	3.88 (0.73)	−10.754 (264.298), <0.001 *	1.278
Meat consumption social norms	3.39 (0.90)	2.48 (0.82)	8.968 (284), <0.001	1.061
Self-efficacy	4.26 (0.65)	3.46 (0.73)	9.791 (284), <0.001	1.158
Food neophobia	2.05 (0.67)	2.44 (0.70)	−4.785 (284), <0.001	0.566
Environmental awareness	4.16 (0.67)	3.52 (0.77)	7.486 (284), <0.001	0.886
Life satisfaction	7.62 (1.84)	6.34 (2.11)	5.485 (284), <0.001	0.649
Subjective health assessment	4.17 (0.75)	3.70 (0.81)	5.127 (284), <0.001	0.606

* Welch’s *t*-test.

**Table 2 foods-13-03215-t002:** Descriptive Statistics: Chi-square and *t*-test Comparisons between Ethical Flexitarians (*n*_1_ = 140) and Utilitarian Flexitarians (*n*_2_ = 146) Regarding Demographic Characteristics and Trust in Various Information Sources.

Variable	Ethical Flexitarians (*%*)	Utilitarian Flexitarians (*SD*)	*χ^2^ *(*df*), *p*	*Phi*
Gender (*n* = 285), female	78.4	58.2	13.366 (1), <0.001	0.217
Education (*n* = 286), College or University degree	55.7	56.2	0.006 (1), 0.939	0.005
Partnership (*n* = 285), in a relationship	63.6	70.3	1.478 (1), 0.224	0.072
(*n* = 285) Living in a city	57.9	48.6	2.444 (1), 0.118	0.092
	*M* (*SD*)	*M* (*SD*)	*t*(*df*), *p*	*Cohen’s d*
Self-assessment of financial struggles	4.02 (1.07)	3.71 (1.13)	2.371 (284), 0.018	0.281
Self-assessment of social status	5.22 (1.61)	5.36 (1.78)	−0.704 (284), 0.482	0.083
Information—authority sources	3.84 (0.75)	3.59 (0.76)	2.783 (284), 0.006	0.329
Information—personal sources	3.73 (0.73)	3.61 (0.65)	1.375 (284), 0.170	0.163
Information—official sources	2.65 (0.85)	2.69 (0.78)	−0.380 (284), 0.704	0.045
Information—unofficial sources	2.59 (0.75)	2.44 (0.68)	1.717 (284), 0.087	0.203

**Table 3 foods-13-03215-t003:** Food consumption, % of responses “once per month or less” or “never” for Ethical flexitarians (*n*_1_ = 140) and Utilitarian flexitarians (*n*_2_ = 146).

Food Type	Ethical Flexitarians (*%*)	Utilitarian Flexitarians (*SD*)	Fisher Exact Test, *p*	*Phi*
Vegetables	2.9	5.5	0.379	0.065
Fish	88.6	86.3	0.597	0.034
Beef	90.7	80.1	0.012	0.149
Pork	87.1	67.8	<0.001	0.231
Poultry	58.6	30.1	<0.001	0.286
Processed meat (e.g., sausages, pates)	69.3	44.5	<0.001	0.250
Eggs	52.1	40.4	0.058	0.118
Milk and diary	25.0	13.0	0.010	0.153

**Table 4 foods-13-03215-t004:** Reasons for not eating (or eating less) meat, % of responses for Ethical flexitarians and Utilitarian flexitarians.

Food Type/Reason	Ethical Flexitarians (*%*)	Utilitarian Flexitarians (*SD*)	Fisher Exact Test, *p*	*Phi*
Fish (*n* = 127)			<0.001	0.330
Taste/smell	34.8	39.7		
Environment/animal exploiting	30.4	5.2		
Other	34.8	55.2		
Beef (*n* = 123)			<0.001	0.329
Taste/smell	20.2	35.9		
Environment/animal exploiting	58.3	23.1		
Other	21.4	41.0		
Pork (*n* = 143)			<0.001	0.393
Taste/smell	23.2	39.6		
Environment/animal exploiting	50.5	10.4		
Other	26.3	50.0		
Poultry (*n* = 64)			0.005	0.397
Taste/smell	8.3	37.5		
Environment/animal exploiting	77.1	37.1		
Other	14.6	25.0		
Processed meat (*n* = 88)			0.005	
Taste/smell	12.7	20.0		0.332
Environment/animal exploiting	47.6	12.0		
Other	39.7	68.0		

**Table 5 foods-13-03215-t005:** Descriptive Statistics: *t*-Test Comparisons between Ethical Flexitarians (*n*_1_ = 140) and Utilitarian Flexitarians (*n*_2_ = 146) Regarding Attitudes Toward Reducing Meat Consumption To a Maximum Of Once a Week.

Attitude	Ethical Flexitarians *M (SD)*	Utilitarian Flexitarians *M* (*SD*)	*t* (*df*), *p*	Cohen’s *d*
Bad (1)—good (5)	4.51 (0.98)	3.36 (1.11)	9.271 (282.127), <0.001 *	1.094
Harmful (1)—useful (5)	4.47 (1.00)	3.39 (1.17)	8.433 (280.595), <0.001 *	0.994
Unpleasant (1)—pleasant (5)	4.27 (1.03)	2.92 (1.10)	10.695 (284), <0.001	1.265
Unsatisfactory (1)—satisfactory (5)	4.28 (1.08)	3.05 (1.03)	9.813 (284), <0.001	1.161
Stupid (1)—smart (5)	4.38 (1.06)	3.44 (1.13)	7.234 (284), <0.001	0.856
Impossible (1)—possible (5)	4.37 (1.16)	3.45 (1.18)	6.710 (284), <0.001	0.794

* Welch’s *t*-test.

**Table 6 foods-13-03215-t006:** Knowledge of and experience with ALT for Ethical flexitarians (*n* = 140) and Utilitarian flexitarians (*n* = 146).

Knowledge/Experience	Ethical Flexitarians (%)	Utilitarian Flexitarians (*SD*)	Fisher Exact Test, *p*	*Phi*
Know cultivated meat	45.7	35.6	5.14, 0.075	0.134
Would try cultured meat	50.0	44.5	3.17, 0.205	0.105
Know plant based meat	57.1	31.5	19.14, <0.001	0.258
Know insects	27.9	28.8.	1.75, 0.424	0.078
Would try insects	14.3	24.0	4.31, 0.121	0.123
Have tried insects	5.0	8.2	1.23, 0.558	0.066

**Table 7 foods-13-03215-t007:** Attitudes towards plant-based meat for Ethical flexitarians (*n*_1_ = 140) and Utilitarian flexitarians (*n*_2_ = 146).

Attitude	Ethical Flexitarians *M* (*SD*)	Utilitarian Flexitarians *M* (*SD*)	*t* (*df*), *p*	Cohen’s *d*
Unhealthy (1)—healthy(5)	3.91 (1.06)	3.39 (0.99)	4.326 (284), <0.001	0.512
Unnatural (1)—natural (5)	3.67 (1.23)	2.82 (1.21)	5.890 (284), <0.001	0.697
Bad for environment (1)—good for environment (5)	3.88 (1.06)	3.32 (1.09)	4.374 (284), <0.001	0.517
Unethical (1)—ethical (5)	4.13 (1.07)	3.51 (1.03)	4.942 (284), <0.001	0.585
Unattractive (1)—attractive (5)	3.74 (1.17)	2.76 (1.14)	7.187 (284), <0.001	0.850
Untasty (1)—tasty (5)	3.74 (1.11)	2.82 (1.02)	7.236 (279.664), <0.001 *	0.857
Not safe for eating (1)—safe for eating (5)	3.94 (1.11)	3.26 (0.99)	5.491 (284), <0.001	0.650
Expensive (1)—cheap (5)	2.58 (1.16)	2.25 (1.11)	2.475 (284), 0.014	0.293
Bad for animals (1)—good for animals (5)	4.13 (1.11)	3.69 (1.11)	3.332 (284), <0.001	0.394
Innutritious (1)—nutritious (5)	3.94 (1.10)	3.04 (1.04)	7.079 (284), <0.001	0.837
Not needed (1)—needed (5)	3.84 (1.22)	3.16 (1.13)	4.889 (284), <0.001	0.578
Bad (1)—good (5)	3.94 (1.08)	3.12 (1.01)	6.639 (284), <0.001	0.785
Aversive (1)—delicious (5)	3.71 (1.08)	2.88 (0.92)	6.963 (273.185), <0.001 *	0.826
Unavailable (1)—available (5)	3.49 (1.20)	2.76 (1.12)	5.291 (284), <0.001	0.626

* Welch’s *t*-test.

**Table 8 foods-13-03215-t008:** Attitudes towards cultured meat for Ethical flexitarians (*n*_1_ = 140) and Utilitarian flexitarians (*n*_2_ = 146).

Attitude	Ethical Flexitarians *M* (*SD*)	Utilitarian Flexitarians *M* (*SD*)	*t* (*df*), *p*	Cohen’s *d*
Unhealthy (1)—healthy(5)	2.96 (1.25)	2.78 (0.98)	1.379 (263.401), 0.169 *	0.164
Unnatural (1)—natural (5)	2.32 (1.38)	2.03 (1.17)	1.898 (272.130), 0.059 *	0.225
Bad for environment (1)—good for environment (5)	3.44 (1.31)	3.14 (1.11)	2.031 (272.301), 0.043 *	0.241
Unethical (1)—ethical (5)	3.25 (1.33)	3.23 (1.17)	0.162 (284), 0.872	0.019
Unattractive (1)—attractive (5)	2.69 (1.34)	2.52 (1.16)	1.113 (275.198), 0.267 *	0.132
Untasty (1)—tasty (5)	2.97 (1.24)	2.82 (0.98)	1.178 (264.723), 0.240 *	0.140
Not safe for eating (1)—safe for eating (5)	3.17 (1.28)	2.83 (1.09)	2.437 (272.486), 0.015 *	0.289
Expensive (1)—cheap (5)	2.27 (1.25)	2.16 (1.20)	0.786 (284), 0.432	0.093
Bad for animals (1)—good for animals (5)	3.79 (1.26)	3.40 (1.19)	2.737 (284), 0.007	0.324
Innutritious (1)—nutritious (5)	3.39 (1.31)	2.8 (1.12)	3.72 (273.345), <0.001 *	0.442
Not needed (1)—needed (5)	3.21 (1.52)	2.97 (1.15)	1.509 (259.146), 0.133 *	0.179
Bad (1)—good (5)	3.25 (1.40)	2.82 (1.00)	2.964 (250.976), 0.003 *	0.353
Aversive (1)—delicious (5)	2.95 (1.20)	2.66 (0.86)	2.314 (251.299), 0.022 *	0.276

* Welch’s *t*-test.

**Table 9 foods-13-03215-t009:** Attitudes towards eating insects for Ethical flexitarians (*n*_1_ = 140) and Utilitarian flexitarians (*n*_2_ = 146).

Attitude	Ethical Flexitarians *M* (*SD*)	Utilitarian Flexitarians *M* (*SD*)	*t* (*df*), *p*	Cohen’s *d*
Unhealthy (1)—healthy (5)	3.19 (1.32)	2.99 (1.11)	1.337 (271.398), 0.182*	0.159
Unnatural (1)—natural (5)	3.27 (1.41)	3.14 (1.31)	0.834 (284), 0.405	0.099
Bad for environment (1)—good for environment (5)	3.09 (1.30)	3.18 (1.07)	−0.653 (269.576), 0.514 *	−0.078
Unethical (1)—ethical (5)	2.74 (1.31)	2.92 (1.00)	−1.371 (259.924), 0.172 *	−0.163
Unattractive (1)—attractive (5)	1.89 (1.15)	1.90 (1.08)	−0.033 (284), 0.973	−0.004
Untasty (1)—tasty (5)	2.43 (1.15)	2.47 (1.05)	−0.339 (284), 0.735	−0.040
Not safe for eating (1)—safe for eating (5)	3.02 (1.33)	2.87 (1.08)	1.055 (268.446), 0.292 *	0.125
Expensive (1)—cheap (5)	2.62 (1.07)	2.51 (1.02)	0.928 (284), 0.354	0.110
Innutritious (1)—nutritious (5)	3.40 (1.31)	3.25 (1.13)	1.014 (274.056), 0.311 *	0.120
Unnecessary (1)—necessary (5)	2.67 (1.32)	2.69 (1.15)	−0.139 (284), 0.889	−0.016
Bad (1)—good (5)	2.74 (1.14)	2.65 (0.97)	0.681 (284), 0.496	0.081
Aversive (1)—delicious (5)	2.30 (1.15)	2.13 (0.96)	1.352 (271.371), 0.117 *	0.161
Unavailable (1)—available (5)	2.52 (1.04)	2.37 (0.93)	1.302 (284), 0.194	0.154
Unhygienic (1)—hygienic (5)	2.54 (1.24)	2.54 (1.03)	−0.040 (270.204), 0.968 *	−0.005

* Welch’s *t*-test.

## Data Availability

The data presented in this study are available on request from the corresponding author. The data are not publicly available due to privacy restrictions.

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
