# Peer review of "Changing Our Food Habits One Bite at a Time: Exploring Young Flexitarians in a Country with a High Meat Intake"

_foods, 2024, doi:10.3390/foods13203215_

Round 1

Reviewer 1 Report

Comments and Suggestions for Authors

The following statement (rows 99-101) should be refined and/or put into context, as the EAT-Lancet Diet was strongly challenged so it cannot be the only benchmark.

Author Response

Comments 1: The following statement (rows 99-101) should be refined and/or put into context, as the EAT-Lancet Diet was strongly challenged so it cannot be the only benchmark.

Response 1:

Indeed EAT-Lancet diet has been praised and challenged many times. Thank you for pointing this out. In order not to add to the length of the article, we removed this reference and included one that provides a better contextualized reference on nutritional issues in Slovenia by Fras et al (under the reference 36). (in row 108)

Reviewer 2 Report

Comments and Suggestions for Authors

The manuscript Saving Our Food Systems One Bite at a Time: Exploring Young Felixtarians in a Country with a High Meat Intake addresses a very important issue. The manuscript is well written, however, in many sections, information is unnecessarily repeated.

Regarding the methodology, using principal components is a good tool to reduce the dimensionality of the variables included in the segmentation analysis, however, some of these variables only included 3 items and only one was eliminated by this procedure. In addition, in my opinion, if by using only one principal component the variability only explained less than 60% of the variation an additional component should be included to increase the explanation of the variability of the variables Food Neophobia and Environmental awareness.  

I recommend making a more concise discussion of the results and especially more concise conclusions. Conclusions should be made based on the results and discussion of the study. Also, I recommend reducing the title keeping only the second part. Although the first part of the title is attractive, the study and the manuscript don't address this issue. (saving the food systems one bite at a time) 

Author Response

Thank you for reviewing the manuscript, good observations and suggestions for improvements of the manuscript. Please find the detailed responses below and the corresponding revisions/corrections highlighted/in track changes in the re-submitted files.

Comments 1: [The manuscript Saving Our Food Systems One Bite at a Time: Exploring Young Felixtarians in a Country with a High Meat Intake addresses a very important issue. The manuscript is well written, however, in many sections, information is unnecessarily repeated..]

Response 1: Thank you for noting this and encouraged us to revise the manuscript and delete the sentences that could be omitted without affecting the main argument. 

We have deleted and accordingly adopted parts in the following rows across sections (visible in the manuscript with track changes):
-    71, 78, 80, 104, 106, 156, 330, 537, 542, 544, 556, 651, 741, 
-    The conclusion was shortened and restructured (rows 768-816).

Comments 2: [Regarding the methodology, using principal components is a good tool to reduce the dimensionality of the variables included in the segmentation analysis, however, some of these variables only included 3 items and only one was eliminated by this procedure. In addition, in my opinion, if by using only one principal component the variability only explained less than 60% of the variation an additional component should be included to increase the explanation of the variability of the variables Food Neophobia and Environmental awareness.]

Response 2: Thank you for this valuable feedback. While we acknowledge the limitation of having a small proportion of variance explained by using only one extracted component, we are concerned that our PCA results do not support the extraction of two components. In both cases—Food Neophobia and Environmental Awareness—the removed items do not form a unique component with a clearly distinguishable meaning, unlike the first extracted component. Instead, they appear scattered with low loadings and low reliability, even after applying a rotated solution. We have now included this observation in the study's limitations.

The added text under the limitation section in the manuscript is (rows 764-768): 
Fourthly, there are methodological concerns regarding the limited variance explained by a single extracted component, especially in the contexts of Food Neophobia and Environmental Awareness. Unfortunately, our findings did not support the extraction of multiple components that would substantially enhance the total explained variance.

Comments 3: [I recommend making a more concise discussion of the results and especially more concise conclusions. Conclusions should be made based on the results and discussion of the study.]

Response 3: We agree with your recommendation and aimed to improve this. We modified both discussion and conclusions.

The conclusion was shortened and restructured as follows (rows 773-926):

Even young people today, who have the opportunity to choose between different dietary patterns, often continue to adhere to the established dietary practices that they learned in their primary social environment. The majority of young adults in Slovenia are not necessarily aware of the correlation between health and environmental issues associated with meat consumption, may deny the existence of problems related to meat consumption, and/or are not motivated to address the issue by modifying their dietary habits and reducing their meat consumption. A considerable number of respondents nevertheless expressed openness to the idea of reducing their meat intake and indicate a potential for a future shift in attitudes towards meat consumption. A noteworthy proportion of young adults indicated their intention to reduce their meat intake. 

Given Slovenia’s historical tradition of promoting meat consumption, it is crucial to raise awareness of the environmental footprint related to the production and consumption of meat in the context of the country’s extensive meat consumption culture. Interventions could concentrate on incremental alterations, underscoring the significance of incremental advancements in the direction of meat reduction. For example, the promotion of "Meatless Mondays" or the encouragement of plant-based substitutions in familiar dishes may be effective strategies. Proper intervention strategies should facilitate the accessibility of plant-based meals in comparison to those that include meat, nudge behaviours in grocery shops and canteens in educational or employment facilities. Moreover, social norms (instrumentalized via peer-led initiatives, social media campaigns, and community events) may be efficacious to emphasise that a considerable proportion of one's social circle is already reducing meat consumption. This may encourage flexitarians to maintain their behaviour and to persuade others to follow suit.

The recognition of several social, psychological, environmental, and practical factors could facilitate flexitarians to maintain their changed behaviour and to enhance its impact. The availability (and affordability) of plant-based alternatives is a significant factor. Those who adopt a flexitarian diet for environmental or ethical reasons may be more likely to maintain their dietary change, less likely to revert to a diet high in meat, and more likely to continue reducing their meat intake. The exposure to educational campaigns or documentaries has the potential to reinforce this commitment. In contrast, Utilitarian flexitarians require social recognition of their dietary choices and additional encouragement to maintain or reduce their meat consumption. Furthermore, they benefit from the convenience and affordability of plant-based and alternative proteins.

Addressing food neophobia and resistance to trying new foods is of paramount importance for individuals across the behavioural spectrum, including flexitarians. It would be beneficial for interventions to educate young adults about alternative protein sources, cooking techniques, and the variety of plant-based options that are currently available.”

Comments 4: [Also, I recommend reducing the title keeping only the second part. Although the first part of the title is attractive, the study and the manuscript don't address this issue. (saving the food systems one bite at a time) ]

Response 4: Thank you for your insightful observation. The current title aims to highlight the critical role of consumer support in transforming unsustainable food systems. It underscores the necessity for consumers to recognize and harness their individual power to drive systemic change. However, if the reviewer finds the first part of the title misleading, we propose the following alternative:
Changing Our Food Habits One Bite at a Time: Exploring Young Flexitarians in a Country with a High Meat Intake

Reviewer 3 Report

Comments and Suggestions for Authors

The study conducted by Kamin et al. is relevant and presents innovative and emergent results. However, I would appreciate the authors could address the following issues:

The main results (values with statistical significance) should be mentioned in the abstract.

References have to be formatted according to the journal’s guidelines.

The Introduction section is too long, please make it more concise.

In the Methods section, you need to explain/justify the calculation of your sample size.

Did you make a pre-test of the employed survey? Was it validated for the study population?

Some parts of your Conclusions should be moved to the Discussion section and make the Conclusions more specific and concise.

Author Response

Thank you for reviewing the manuscript, good observations and suggestions for improvements of the manuscript. Please find the detailed responses below and the corresponding revisions/corrections highlighted/in track changes in the re-submitted files. 

Comments 1: [The main results (values with statistical significance) should be mentioned in the abstract]

Response 1: Thank you for suggesting this. We added the suggested information in the abstract.

We included the following text in the abstract: 
Utilitarian flexitarians exhibit a stronger preference for meat (t(284) = -15.180, p < 0.001), greater food neophobia (t(284) = -4.785, p < 0.001), and lower environmental awareness (t(284) = 7.486, p < 0.001) compared to Ethical flexitarians. The Ethical group, predominantly female (χ²(1) = 13.366, p < 0.001), demonstrates higher life satisfaction (t(284) = 5.485, p < 0.001), better health perceptions (t(284) = 5.127, p < 0.001), and stronger beliefs in reducing meat consumption (t(284) = -8.968, p < 0.001). Additionally, Ethical flexitarians hold more positive views on plant-based meat, perceiving it as healthier (t(284) = 4.326, p < 0.001) and more ethical (t(284) = 4.942, p < 0.001), and show a greater willingness to adopt it (t(284) = 7.632, p < 0.001). While both groups possess similar knowledge and willingness regarding cultured meat and insects, Ethical flexitarians view cultured meat more favorably (t(250.976) = 2.964, p = 0.003).”

Comments 2: [References have to be formatted according to the journal’s guidelines.]

Response 2: Thank you for your observation. Indeed, there were instances where the surnames of authors were retained in the text. These have now been completely removed, and only numerical references are maintained throughout the document.  

Comments 3: [The Introduction section is too long, please make it more concise]

Response 3: Thank you for your suggestion to revise the introduction and eliminate the overly pedagogical approach. We have now removed certain sections that were not essential to the main argument. We believe the introduction is now more concise and easier to follow. The changes are visible in the revised manuscript with track changes function.

Comments 4: [In the Methods section, you need to explain/justify the calculation of your sample size. ]

Response 4: Thank you for your comment. We have now added a short explanation about the decision on the sample size in the method section.

We added the following explanation (rows 189-200): 
We aimed to obtain the largest quasi-representative sample possible within our financial constraints, considering the studied phenomenon and sample sizes in previous similar studies. Our sample was sufficient to conduct the desired tests and measure the studied phenomena in the selected population for the first time. 

Comments 5: [Did you make a pre-test of the employed survey? Was it validated for the study population?]

Response 5: Thank you for your question. Before sending the questionnaire to Valicon, who conducted the study on their panel, we reviewed it with experts and the target population. The revised questionnaire was then sent to Valicon, where it was tested on a sample from their panel before being administered to the final study sample. 

We added the following explanation (rows 200-201): 
The questionnaire was reviewed with experts and the target population before it was administered to the final study sample. 

Comments 6: [Some parts of your Conclusions should be moved to the Discussion section and make the Conclusions more specific and concise.]

Response 6: Thank you for sharing your observation. The conclusion was shortened and restructured as follows (rows 773-926):

“Even young people today, who have the opportunity to choose between different dietary patterns, often continue to adhere to the established dietary practices that they learned in their primary social environment. The majority of young adults in Slovenia are not necessarily aware of the correlation between health and environmental issues associated with meat consumption, may deny the existence of problems related to meat consumption, and/or are not motivated to address the issue by modifying their dietary habits and reducing their meat consumption. A considerable number of respondents nevertheless expressed openness to the idea of reducing their meat intake and indicate a potential for a future shift in attitudes towards meat consumption. A noteworthy proportion of young adults indicated their intention to reduce their meat intake. 

Given Slovenia’s historical tradition of promoting meat consumption, it is crucial to raise awareness of the environmental footprint related to the production and consumption of meat in the context of the country’s extensive meat consumption culture. Interventions could concentrate on incremental alterations, underscoring the significance of incremental advancements in the direction of meat reduction. For example, the promotion of "Meatless Mondays" or the encouragement of plant-based substitutions in familiar dishes may be effective strategies. Proper intervention strategies should facilitate the accessibility of plant-based meals in comparison to those that include meat, nudge behaviours in grocery shops and canteens in educational or employment facilities. Moreover, social norms (instrumentalized via peer-led initiatives, social media campaigns, and community events) may be efficacious to emphasise that a considerable proportion of one's social circle is already reducing meat consumption. This may encourage flexitarians to maintain their behaviour and to persuade others to follow suit.

The recognition of several social, psychological, environmental, and practical factors could facilitate flexitarians to maintain their changed behaviour and to enhance its impact. The availability (and affordability) of plant-based alternatives is a significant factor. Those who adopt a flexitarian diet for environmental or ethical reasons may be more likely to maintain their dietary change, less likely to revert to a diet high in meat, and more likely to continue reducing their meat intake. The exposure to educational campaigns or documentaries has the potential to reinforce this commitment. In contrast, Utilitarian flexitarians require social recognition of their dietary choices and additional encouragement to maintain or reduce their meat consumption. Furthermore, they benefit from the convenience and affordability of plant-based and alternative proteins.

Addressing food neophobia and resistance to trying new foods is of paramount importance for individuals across the behavioural spectrum, including flexitarians. It would be beneficial for interventions to educate young adults about alternative protein sources, cooking techniques, and the variety of plant-based options that are currently available.”